# Exploring Psychological Needs and Burden of Care in Parents of Children with Hemato-Oncological Diseases

Loredana Benedetto [1] , Irene Marino [2], Francesca Ronco [3], Grazia Iaria [3], Luisa Foletti [4] and Massimo Ingrassia [1,*]

[1] Department of Clinical and Experimental Medicine, University of Messina, via Consolare Valeria 1, I-98125 Messina, Italy; loredana.benedetto@unime.it

[2] Pediatric and Clinical Psychologist, via Germano Sommeiller 25, I-00185 Roma, Italy; irene.marino@live.it

[3] Pediatric Onco-Hematology Unit, "Bianchi-Melacrino" Hospital, via Petrara 1, I-89123 Reggio Calabria, Italy; francescaronco29@gmail.com (F.R.); ematologiarc@alice.it (G.I.)

[4] AIL (Associazione Italiana Contro le Leucemie-Linfomi e Mieloma), Section "A. Neri", via Aurora 1, I-89124 Reggio Calabria, Italy; luisellafoletit@gmail.com

*  Correspondence: massimo.ingrassia@unime.it

**Abstract:** Caring for a child with an acute/life threatening disease exposes parents to multiple stressors and challenges, resulting in a physical and psychological burden. Parents experience many health-related issues and worries that often remain underestimated. The aims of the study were: (a) to explore the associations between needs/disease-related issues and burden in parents of children with leukemia or Hodgkin's disease; (b) to estimate predictors of parents' burden using a stepwise linear regression analysis. Children (N = 33) followed an active therapy protocol (48.5%), or they were off therapy (51.5%). Forty-four parents completed surveys on caregiver burden levels and needs to cope with the child's illness. Parental factors impacting burden (personal resources, loss of control, depression) and child's quality of life (QoL) were also assessed. Among the needs, information about the illness/resources were the most urgently expressed by parents, followed by reassurance against fears for the child's development and future well-being. Parents reported severe (27.3%) and moderate (22.7%) burden, with a higher percentage of caregivers with severe burden in the off-therapy phase (18.2%) than in the active-therapy phase (9.1%). The child's decreased physical QoL and parent's loss of control predicted higher levels of burden. The implications for supportive interventions aimed at responding to parental needs and preventing caregiver burden are discussed.

**Keywords:** caregiver burden; parents; needs; pediatric hemato-oncological disease

## 1. Introduction

Despite the therapeutic advances that have positively changed the prognosis of pediatric hemato-oncological diseases [1,2], cancer remains a "life-changing experience" [3] for parents. Leukemia is characterized by an abnormal hematopoietic tissue, and it is the most common malignant disease in children, with an age-standard incidence of 5 cases per 100,000 (0–19 years old) in Italy (3.2 per 100,000 worldwide [2]). As an acute disease, particularly in the form of acute lymphoblastic leukemia [4], its diagnosis causes emotional suffering, helplessness for the rapid course of illness, and worries for worsening of child's health and life [5] among parents. Parents experience anxiety for the child's hospitalizations, increased distress [6], and uncertainty for cycles of medication that are invasive and long-lasting (usually, over 2–3 years of treatment [4]). In addition, parents must reorganize their family role and routines across the course of child's illness, facing difficulties that threatens their physical and psychological health [7]. In the psychological literature, these negative consequences resulting from caregiving have been described as *burden of care*, that is [8] (p. 69) "the degree to which the caregiver perceives that the different spheres of his/her life (social life, leisure, health, privacy) have been affected [. . . ]" by disease-related primary care

tasks. Parents play a critical role in the child's adjustment to illness and are highly involved in all therapeutic phases. Caregiver burden identifies multiple challenges, responsibilities, and stressors which add to routine parental tasks when children have chronic or severe illnesses, such as cerebral palsy or brain tumors [8–10]. However, studies investigating caregiver burden in pediatric hemato-oncological diseases are still scarce [8,11,12].

Caregiver burden is a multidimensional construct [9,13], including objective factors (such as time for assistance or financial resources) and subjective factors (physical, emotional, social). Caring for a child with leukemia forces the parent (usually, the mother) to "switch off" from everything else to focus on the child's physical and emotional needs during hospitalization, transplantation, or chemotherapy. *Time for care* is a determining factor in the perception of burden: particularly, caring for a child with cancer requires hospitalizations and constant vigilance, especially because the parent may feel responsible for the child's pain or health conditions [14]. *Physical burden* includes fatigue, insomnia, decreased appetite, headache, and anxiety symptoms associated with worries and constant changes in the parent's life [15]. *Emotive burden* describes negative emotions (anger, embarrassment, helplessness, etc.) towards the child and his/her behaviors. Children often react to physical pain and limitations due to hospitalizations with fear, irritability, refusal of treatment, or withdrawal that carries over into the relationship with the parent, usually the mother [16]. A further source of emotional concern arises from other children who are neglected and suffer from the physical absence of the parents involved in the assistance of the sick child [17]. *Social burden* describes the perception of restriction and inner conflict between the roles that the person is called upon to play in the workplace, family, or social contexts. Familial conflicts, decline in marital satisfaction, and clinical levels of couple distress have been observed [18]. Regardless of the child's age, the needs for assistance and care during the long treatment (repeated hospitalizations, chemotherapy, or checks) force the parent to limit social contacts, in turn reducing the potential sources of support [19]. Conversely, social support can be a factor that moderates the parental caregiving burden [20]. Finally, *Financial burden* has recently been investigated among the long-term consequences of cancer. Economic difficulties can derive from parent's absences and pressures from work, travels, and care expenses. It has been observed that these difficulties are not limited to the period of treatment, because 1–5 years after diagnosis, the expenses that parents face can still be greater than the pre-morbid period [21]. Thereby, this economic burden, joined to worries about the future and low family quality of life (QoL), increases the overall overload perceived by parents [22].

The assessment of psychological needs and diseases-related problems associated with parental experiences with cancer is important for capturing factors impacting caregiver burden, but empirical studies are still lacking [23]. Parental needs and challenges change along the course of the child's illness, starting from emotional and informative support that reduces confusion and fear at diagnosis, progressing through other issues involving the parents, such as the management of child's pain or therapy (transplantation, chemotherapy, or radiation), the relationships with healthcare professionals, coping with stress and family organization, up to and including the spiritual needs of the parent and the fear of death [24]. Parents interpret the child's illness and his/her actual QoL, the disease-related problems to be faced, and the available resources in relation to caregiving. All these factors can be conceptualized as a set of variables influencing the process of burden [9,25]. Based on these premises, the aims of current study were: (a) to explore the psychological needs and issues experienced by parents of children with hemato-oncological diseases; (b) to assess the associations between these factors and the dimensions of caregiver burden; and finally, to verify if the risk for burden changes as a function of (c) the parent's adaptation (personal resources, loss of control, depression) and (d) the child's health-related QoL.

Moving from the stress-coping theory of Lazarus and Folkman [26], van der Borne and colleagues [27] explored the psychosocial problems influencing distress in parents of children with a chronic/life threatening illness. Based on these theoretical assumptions, the current study assessed the individual characteristics that could influence burden levels.

Particularly, parent's negative feelings (fear and depression [8]), loss of control, and threats to self-esteem have been linked to higher psychological distress, worry about the child's future, and ineffective coping with the child's disease [8,28]. The hypothesis to be tested is that these determinants negatively impact the parents' burden of caring. Finally, the choice of introducing an evaluation of the child's QoL is based on previous studies supporting the link between the child's cancer-related factors (e.g., activity limitation, type of active treatment, etc.), parental stress, and low QoL [29], and between the reported poor QoL of the child and maternal depression [30]. However, the association between children's health-related QoL and burden experienced by parents in the context of hemato-oncological diseases has been largely unexplored. Therefore, another hypothesis to be tested is that parenting burden levels could be significantly associated with and explained by the child's decreased health-related QoL.

## 2. Methods

### 2.1. Participants and Procedure

The recruitment of participants took place in the Pediatric Hemato-Oncology Unit of a public hospital in Reggio Calabria, Italy. The inclusion criteria were: being a parent of a child under age 18 with a hemato-oncological disease, and a native Italian speaker. Parent's current mental illness and child's additional physical illness (e.g., diabetes mellitus, chronic heart/pulmonary disease, etc.) were exclusion criteria.

Parents were invited to participate during their child's routine hospital visits. Before enrollment, parents received information about study aims and signed the informed consent. Then, in a reserved room of the day hospital, they completed two self-reporting instruments (i.e., the *Caregiver Burden Inventory* and the *Psychological Needs and Disease-Related Issues* questionnaire), and a health-related QoL survey (KINDL$^R$) for their child, if the child was younger than 7 years old. Children 7 years old and older personally answered the health-related QoL questions using the self-reporting version of KINDL$^R$. All invited parents ($N = 44$) and their children ($N = 33$) participated. Specifically, the parents were 19 mothers and 3 fathers participating individually, and 11 father–mother pairs.

### 2.2. Measures

CBI—*Caregiver Burden Inventory*, Italian adaptation [31,32]: 22 questions answered on a Likert scale from 0 (not at all descriptive) to 4 (very descriptive) measured caregiver's experience on five dimensions: *Time-dependence* (restrictions on the time available), *Developmental burden* (feeling of not being able to enjoy one's life like peers, but instead, always being anxious and tense), *Physical* (chronic fatigue and physical symptoms), *Social* (role conflict, such as work-family, or discussing with the spouse or other members regarding how to manage the patient's needs), and *Emotional burden* (negative feelings about the family member who needs care). Higher scores correspond to greater burden. The total scores serve as basis to distinguish three levels of parent burden: not problematic (CBI $\leq$ 24), moderate (24 < CBI < 36), and severe (CBI $\geq$ 36). Particularly, the total score makes it possible to distinguish a serious caregiver burnout condition (total CBI $\geq$ 36) from the person's need to be supported or replaced in assistance (36 > CBI > 24). In the present sample, the CBI reliability was the same as for the Italian adaptation [32]: Cronbach's alpha = 0.95.

N-DRI—*psychological Needs and Disease-Related Issues* questionnaire: Questions are derived from van Der Borne et al.'s study [27] assessing parents' needs and strengths to cope with the illness of their child. The Italian adaptation [33], already used in a previous study with children with chronic and/or life-limiting diseases [34], measures: *Uncertainly about disease* (11 items; e.g., "Quite a lot/very much need of information about: How my child can develop") and *resources* (8 items; e.g., "How I can talk to or deal with the doctor"); *Fears for the child* (4 items; e.g., "Quite/very much concerned about: Disappointments my child has to cope with the future") and *siblings* (4 items added by Ingrassia et al., [34]; e.g., "Other children will feel neglected by me due to the attention I give to this sick

child"); *Loss of control* (12 items; e.g., "Applies to me partly/entirely: My thoughts often wander to concern about my child"); *Fears of themselves* (10 items; e.g., "Quite/very much concerned about: Coming to the end of my patience in assisting my child"), *Depression* (10 items, e.g., "Often/very or always the case: I feel dejected"); *Personal resources* (12 items; e.g., "I agree partly/completely that: I am not easily discouraged"). All statements must be rated on a 4-point Likert scale (from 1 to 4). Higher scores indicate more intensely perceived psychological needs and disease-related issues, with an inverse direction for *Personal resources*. In the present sample, Cronbach's alphas ranged from adequate (*Fears for the child* = 0.68) to excellent (*Uncertainly about disease* = 0.94).

KINDL^R—*Quality of Life questionnaire for children* [35]: Consists of 24 items measuring six dimensions of health-related QoL (i.e., *Physical well-being*, *Emotional well-being*, *Self-esteem*, *Family*, *Friends*, and *School*). The answers are expressed on a 5-point Likert scale (1 = never, to 5 = always), with higher scores indicating better general QoL. Different age-specific versions of KINDL^R are available at the official website (https://www.kindl.org/; accessed on 20 March 20 2020). Here, the parent-reported questionnaire for children (3–6 y.o.) and the self-report questionnaire for children/adolescents (up to 18 y.o.) were used. All versions contain a "*Disease*" subscale that measures the child's disease-related QoL. In the present sample, Cronbach's alphas ranged similarly to the original version (from *School* = 0.63 to *Disease module* = 0.72).

*2.3. Statistical Analysis*

Data were processed with IBM SPSS Statistics for Windows 19.0. Firstly, we calculated descriptive statistics and proportions of the participants demographics and characteristics (i.e., parent age, gender, and marital status; child's age, gender, diagnosis, time since diagnosis—<5 months vs. >5 months—and therapeutic phase, namely, active therapy vs. off therapy—the active therapy condition consisted of chemotherapy, induction, or maintenance treatments). Then, as a preliminary analysis to search for confounding factors, we performed different MANOVAs with parents' measures (i.e., CBI and N-DRI subscales) and children's KINDL^R dimensions as dependent variables, estimating differences as a function of therapeutic phases (active therapy vs. off therapy). Moreover, the proportional differences of caregiver burden levels (not problematic, moderate, and severe) were analyzed through a chi-square test applied to a crosstab 2 (therapeutic phase) × 3 (CBI level).

Secondly, we performed a correlational analysis (Pearson's *r*) to explore the relationships between participants' measures. Then, we estimated the predictors of parental burden using a linear stepwise regression analysis.

Finally, using different MANOVAs with parental measures (i.e., CBI and N-DRI dimensions) as dependent variables we explored the differences between fathers and mothers, even though this estimate was not one of the explicit aims of the study.

**3. Results**

Table 1 summarizes parents' and children's characteristics.

Preliminarily, no differences between parent measures (i.e., CBI and N-DRI) as a function of therapeutic conditions (active therapy [AT] vs. off therapy [OT]) emerged, with only one exception: N-DRI *Uncertainly about resources* resulted higher for AT condition ($M_{AT}$ = 2.94, $SD_{AT}$ = 0.80, vs. $M_{OT}$ = 2.21, $SD_{OT}$ = 0.91; $F(1, 42)$ = 7.95, $p$ = 0.007, $\eta_p^2$ = 0.16). Additionally, child's KINDL^R means did not differ as a function of therapeutic conditions. Then, a chi-square test applied to a cross tab 2 (AT vs. OT) × 3 (levels of burden) revealed the non-independence between factors ($\chi^2(2)$ = 8.46, $p$ = 0.015).

**Table 1.** Detected features of parents and children.

| Parent Characteristics [1] (N = 44) | | |
|---|---|---|
| Age (years) | *M* = 40.3 (range 29–53) | |
| Gender | | |
|   Female | 30 (72.3%) | |
|   Male | 14 (27.7%) | |
| Marital status | | |
|   Married | 41 (93.2%) | |
|   Divorced/widowed | 3 (6.8%) | |
| Total burden | | |
| | Active therapy | Off therapy |
|   Not problematic | 9 (20.5%) | 13 (29.5%) |
|   Moderate | 9 (20.5%) | 1 (2.2%) |
|   Severe | 4 (9.1%) | 8 (18.2%) |
| Child characteristics [1] (N = 33) | | |
| Age (years) | *M* = 8.3 *(range 3–17)* | |
| Gender | | |
|   Female | 15 (45.5%) | |
|   Male | 18 (54.5%) | |
| Diagnosis [2] | | |
|   ALL | 27 (81.8%) | |
|   HL | 6 (18.2%) | |
| Time since diagnosis | | |
|   <5 months | 7 (21.2%) | |
|   >5 months | 26 (78.8%) | |
| Therapeutic phase | | |
|   Active therapy [3] | 16 (48.5%) | |
|   Off therapy | 17 (51.5%) | |

[1] Scores are expressed as average (range) or frequency (percentage). [2] ALL = acute lymphoblastic leukemia; HL = Hodgkin's lymphoma. [3] chemotherapy, induction, or maintenance treatment.

Subsequently, the correlational analysis revealed several associations between CBI and N-DRI scales with $r > 0.30$ coefficients (see Table 2 for an overview). Actually, the *Needs* of N-DRI scales with higher scores were *Uncertainly about disease*, *Uncertainly about resources*, and *Fears for the child*. Yet, total burden was associated with *Parental adjustment* dimensions (i.e., *Loss of control*, *Depression*, and *Personal resources*) through considerable coefficients (all $rs(44) \geq 0.40$), and was also appreciably associated with *Fears for the child* [$r(44) = 0.37$], but not with *Uncertainly about disease*, and *Uncertainly about resources*. Furthermore, considerable associations emerged between total burden and some child's QoL scales (i.e., *Physical well-being* and *Self-esteem*). Then, the linear stepwise regression analysis revealed, as total burden predictors, a model with two factors, i.e., KINDL$^R$—*Physical well-being* ($\beta = -0.43$; $t = -2.60$, $p = 0.02$) and N-DRI—*Loss of control* ($\beta = 0.39$; $t = 2.35$, $p = 0.03$), explaining 1/4 of variance ($R_c^2 = 0.25$; $F(2, 28) = 5.75$, $p = 0.01$).

Finally, we explored parental gender differences: fathers resulted higher for N-DRI—*Uncertainly about resources* ($M_f = 2.98$, $SD_f = 0.92$, vs. $M_m = 2.38$, $SD_m = 0.87$; $F(1, 42) = 4.46$, $p = 0.04$, $\eta_p^2 = 0.10$), while mothers highlighted more N-DRI—*Depression* ($M_f = 1.46$, $SD_f = 0.68$, vs. $M_m = 1.92$, $SD_m = 0.60$; $F(1, 42) = 5.00$, $p = 0.03$, $\eta_p^2 = 0.11$) and CBI—*Physical burden* ($M_f = 6.43$, $SD_f = 4.58$, vs. $M_m = 3.79$, $SD_m = 3.56$; $F(1, 42) = 3.64$, $p = 0.06$, $\eta_p^2 = 0.08$).

**Table 2.** Study variables descriptive statistics (*M* and *SD*) and related Pearson's correlation coefficients (*r*).

| Measures | | 1. | 2. | 3. | 4. | 5. | 6. | 7. | 8. | 9. | 10. | 11. | 12. | 13. | 14. | 15. | 16. | 17. | 18. | 19. | 20. | 21. |
|---|---|---|---|---|---|---|---|---|---|---|---|---|---|---|---|---|---|---|---|---|---|---|
| | *M* | 9.89 | 5.86 | 5.59 | 0.93 | 3.36 | 25.73 | 2.62 | 2.57 | 2.19 | 1.65 | 1.44 | 2.15 | 1.77 | 3.29 | 15.10 | 14.87 | 13.29 | 13.97 | 13.40 | 9.30 | 18.20 |
| | *SD* | 6.06 | 4.29 | 4.42 | 1.28 | 3.54 | 14.08 | 0.93 | 0.92 | 0.82 | 1.04 | 0.56 | 0.82 | 0.66 | 0.67 | 3.69 | 3.37 | 5.89 | 4.14 | 5.95 | 7.70 | 7.31 |
| *Caregiver burden* | | | | | | | | | | | | | | | | | | | | | | |
| 1. Time-dependence | *r* | 1 | | | | | | | | | | | | | | | | | | | | |
| 2. Developmental burden | *r* | 0.33 * | 1 | | | | | | | | | | | | | | | | | | | |
| 3. Physical burden | *r* | 0.41 ** | 0.74 ** | 1 | | | | | | | | | | | | | | | | | | |
| 4. Emotional burden | *r* | 0.14 | 0.30 * | 0.21 | 1 | | | | | | | | | | | | | | | | | |
| 5. Social burden | *r* | 0.14 | 0.41 ** | 0.48 ** | 0.30 | 1 | | | | | | | | | | | | | | | | |
| 6. Total burden | *r* | 0.70 ** | 0.80 ** | 0.85 ** | 0.38 * | 0.62 ** | 1 | | | | | | | | | | | | | | | |
| *Psychological needs and disease-related issues* | | | | | | | | | | | | | | | | | | | | | | |
| 7. Uncertainly about disease | *r* | 0.30 * | −0.13 | 0.07 | −0.08 | −0.13 | 0.08 | 1 | | | | | | | | | | | | | | |
| 8. Uncertainly about resources | *r* | 0.33 * | −0.17 | 0.01 | 0.05 | 0.00 | 0.11 | 0.69 ** | 1 | | | | | | | | | | | | | |
| 9. Fears for the child | *r* | 0.49 ** | 0.16 | 0.24 | 0.08 | 0.14 | 0.37 * | 0.20 | 0.27 | 1 | | | | | | | | | | | | |
| 10. Fears for the siblings [1] | *r* | 0.04 | 0.08 | 0.10 | 0.14 | 0.14 | 0.12 | 0.12 | 0.11 | 0.33 * | 1 | | | | | | | | | | | |
| 11. Fears for themselves | *r* | 0.32 * | 0.03 | 0.00 | 0.23 | 0.08 | 0.19 | 0.26 | 0.30 * | 0.56 ** | 0.38 * | 1 | | | | | | | | | | |
| *Parental adjustment* [2] | | | | | | | | | | | | | | | | | | | | | | |
| 12. Loss of control | *r* | 0.42 ** | 0.30 * | 0.30 * | 0.19 | 0.34 * | 0.47 ** | 0.22 | −0.01 | 0.40 ** | 0.28 | 0.35 * | 1 | | | | | | | | | |
| 13. Depression | *r* | 0.16 | 0.31 * | 0.42 ** | 0.27 | 0.34 * | 0.40 ** | 0.11 | 0.08 | 0.57 ** | 0.45 ** | 0.45 ** | 0.55 ** | 1 | | | | | | | | |
| 14. Personal resources | *r* | −0.26 | −0.33 * | −0.38 * | −0.06 | −0.37 * | −0.42 ** | 0.12 | 0.00 | −0.42 ** | −0.34 * | −0.38 * | −0.37 * | −0.63 ** | 1 | | | | | | | |
| *Child's Quality of Life* | | | | | | | | | | | | | | | | | | | | | | |
| 15. Physical well-being | *r* | −0.12 | −0.33 | −0.47 ** | −0.10 | −0.38 * | −0.43 * | −0.08 | −0.28 | −0.11 | 0.07 | 0.02 | 0.02 | −0.01 | 0.04 | 1 | | | | | | |
| 16. Emotional well-being | *r* | −0.12 | −0.20 | −0.29 | −0.02 | −0.35 | −0.30 | −0.01 | −0.13 | −0.42 * | −0.10 | −0.17 | 0.04 | −0.13 | 0.26 | 0.64 ** | 1 | | | | | |
| 17. Self-esteem | *r* | −0.37 * | −0.44 * | −0.37 * | −0.08 | −0.09 | −0.44 * | −0.11 | −0.08 | −0.45 * | 0.00 | −0.22 | −0.09 | −0.03 | 0.21 | 0.49 ** | 0.68 ** | 1 | | | | |
| 18. Family | *r* | −0.25 | −0.27 | −0.34 | 0.19 | −0.20 | −0.33 | 0.22 | 0.14 | −0.07 | 0.31 | 0.34 | 0.17 | 0.40 * | −0.03 | 0.45 * | 0.38 * | 0.50 ** | 1 | | | |
| 19. Friends | *r* | −0.19 | −0.26 | −0.18 | −0.14 | −0.17 | −0.28 | 0.04 | 0.05 | −0.26 | −0.01 | −0.00 | −0.03 | 0.09 | −0.09 | 0.38 * | 0.52 ** | 0.57 ** | 0.56 ** | 1 | | |
| 20. School | *r* | −0.09 | −0.10 | 0.03 | 0.23 | −0.00 | −0.04 | 0.01 | 0.07 | 0.06 | 0.11 | 0.17 | −0.16 | 0.28 | −0.12 | 0.23 | 0.11 | 0.24 | 0.48 ** | 0.50 ** | 1 | |
| 21. Disease | *r* | −0.14 | −0.25 | −0.35 | 0.00 | −0.10 | −0.28 | −0.16 | 0.13 | −0.14 | 0.27 | 0.27 | −0.11 | 0.07 | −0.01 | 0.37 * | 0.41 * | 0.45 * | 0.48 ** | 0.25 | 0.30 | 1 |

* $p < 0.05$, two-ties; ** $p < 0.01$, two-ties. [1] Items from Ingrassia et al. [28]; [2] Items from van Der Borne et al. [26].

## 4. Discussion

The current study provides interesting insights about emerging needs among parents of children with hemato-oncological diseases. Receiving appropriate information after diagnosis resulted in the first among unmet needs declared by parents. Mainly, they expressed the need to receive information aimed at reducing the uncertainty about the disease (i.e., effective treatments, prognosis, consequences to the child's physical and psychological health). Furthermore, in the experience of parents, it was urgent to have more resources available to cope with the illness (i.e., communication with health professionals, ways to manage child's physical/emotional symptoms, and practical/emotional support to care for the sick child). Particularly, having more information about resources was expressed as a more urgent need when children were receiving active therapy (vs. off therapy). Therefore, current results suggest to the healthcare team the importance of providing continuous supportive care [24,36], especially over the course of long treatment phases when parents' reassurance and practical needs become more urgent. The third issue that parents mentioned as "quite" relevant was related to worries and fears for the child (future decline, delusions, or restrictions for illness, etc.). This issue is the only one that was associated with higher burden levels, thus confirming that more uncertainty and worries for the child's illness promote parental distress, mainly due to the fear for relapse after treatment and the anguish for his/her survival [37]. In this regard, studies report the occurrence of high rates of post-traumatic stress disorder (PTSD) in parents, suggesting that "the experience of parenting a child with cancer may be more traumatic than the actual disease" [36] (p. 13).

In the current sample, 22.7% of parents reported moderate levels of burden, thus indicating the urgent need to receive practical support, that is, to be replaced in the child's assistance or helped in the organization of family life [37]. Secondly, 27.3% of caregivers reported severe levels of burden that seriously affected their physical, emotional, and social well-being. The percentage of parents with severe burden levels was higher in the off-therapy phase than in the active phase, while the percentage of parents with moderate burden (need for support) decreased in the off-therapy phase. To our knowledge, to date, no study has explored the variations in caregiver burden levels during treatment/off therapy phases in the context of hemato-oncological diseases. A study by Wang and coll. [38] indicated rates of 33.8% of mild-to-moderate burden and 9.2% of severe burden in parents of children newly diagnosed with ALL. In the current study, the percentage of parents experiencing severe burden was 9.1% in the active phase vs. 18.3% in the off-therapy phase, respectively. Unfortunately, the small sample size of patients and the disparity of their clinical characteristics (mainly, time since diagnosis) did not allow for the exploration of the factors that may explain these results. Based on previous empirical evidence, it is known that some parental factors (i.e., more daily care time, low social support and co-parenting, anxiety, and worse general health [9,38]) are predictive of higher caregiver burden levels. However, other longitudinal studies from diagnosis to the conclusion of the cycles of treatments are needed. Findings from the current study are particularly relevant for health professionals and suggest the necessity to plan supportive interventions to reduce parent's burden, not only during active medical therapy, but also when children are off treatment. After all, parents' needs (i.e., worries for sequelae from therapy, future relapse, child's cognitive or social impairment related to illness, etc.) change over time as a function of the child's age [23] and the stage of a child's treatment [30]; therefore, interventions should be modulated to target parenting demands and the experience of the caregiver [23,24].

Based on regression analysis, burden was predicted by the child's QoL related to physical condition, followed by a feeling of loss of control on the part of the parent. In other studies, children with acute leukemia reported the lowest levels of QoL in the domain of physical health, particularly children on therapy [5]; in addition, the severity of symptoms in children with leukemia predicted maternal burden [8]. Taken together, these data are in line with the findings from the current study. Childhood cancer puts parents in a condition of extreme uncertainty and gives them a sense of helplessness due the painful

medical treatments to which the child is subjected [28]. The findings also suggest that low perception of control regarding the child's physical health is linked to the parent's negative psychological outcomes, such as ineffective coping, distress, guilt, depressive symptoms, and decreased life perception [8,29,34,39].

In addition, in the current study, a strong positive correlation emerged between burden and depressive feelings, with mothers reporting significantly higher scores regarding depressive feelings than fathers. Conversely, a coping style focused on cognitive appraisal (i.e., meaning of the illness, spiritual needs, etc.) and problem-solving (i.e., learning about the cancer, exploring treatment options, seeking support, etc.) has been associated with decreased caregiver burden and depressive feelings [10]. Since coping strategies are modifiable factors, these findings highlight the potential benefits of psychological interventions aimed to prevent the development of caregiver burden in parents and its detrimental consequences on the child's and the family's functioning [12,25,40].

This study has some limitations: the first among these is the small number of participants and the predominance of mothers. Some gender differences have been observed, particularly higher levels of fatigue (physical burden) and depressive feelings in mothers. These results are in line with the literature reporting greater maternal vulnerability to burden and family stress [15]. However, in the future, it would be desirable to extend the study and involve a larger number of fathers, with the purpose of exploring how some couple-protective factors, such as co-parenting, communication, perceived support, or dyadic coping [40], can intervene, buffering the caregiver burden. Secondly, within the patient group, there was a disparity in the proportion of children with different acute/life threatening illnesses (81.8% leukemia). Therefore, it was not possible to investigate the impact of leukemia vs. Hodgkin's lymphoma on children's QoL and parent's psychological adjustment. In addition, the present study assessed children's QoL with a generic health-related questionnaire, thus the impact of the severity of clinical symptoms experienced by the child (i.e., fatigue, weakness, or nausea) and the consequences of long-term treatments (mainly, procedural anxiety, hospitalization, activity or school limitation, etc.) were not examined. Future studies could benefit from cancer-specific QoL instruments to corroborate the current study's findings. Finally, the study is cross-sectional and children were separated as a function of active therapy phase vs. off therapy. However, as observed in previous studies [5], the heterogeneity of therapy in the active phase (chemotherapy, induction, or maintenance treatment), together with other clinical characteristics of children, may have affected their health-related QoL. Future longitudinal studies could explore how children's QoL changes along the course of the disease, influencing parental involvement in caring and perceived burden.

## 5. Conclusions

The strengths of the current study are the exploration of the parental needs and factors impacting the burden they experience in coping with a child's hemato-oncological disease. Having identified these influential factors provides insights for interventions. A family-centered approach that includes attention to parents and their unexpressed needs can be derived. Health professionals must recognize that emotional crises and challenges for parents extend beyond the child's hospitalization or the conclusion of chemotherapeutic treatments. Pivotal studies indicate the efficacy of education training intervention in reducing caregiver burden among mothers of children with leukemia [11]. These supportive interventions are promising and appear to be an effective strategy in responding to parents' psychological needs, enhancing resources and mitigating the multiple negative effects of caregiving.

**Author Contributions:** Conceptualization, L.B, I.M. and L.F.; methodology, L.B.; formal analysis, M.I.; investigation, I.M.; resources, F.R., G.I. and L.F.; data curation L.B. and I.M.; writing—original draft preparation, I.M.; writing—review and editing, L.B.; supervision, L.B. and M.I. All authors have read and agreed to the published version of the manuscript.

**Funding:** This research did not receive external funding.

**Institutional Review Board Statement:** The study was conducted in accordance with the Declaration of Helsinki, and approved by the Ethics Committee of the Psychological Research and Intervention Center (CeRIP), University of Messina (protocol code 17972, 14 February 2020).

**Informed Consent Statement:** Informed consent was obtained from all subjects involved in the study.

**Data Availability Statement:** Not applicable.

**Conflicts of Interest:** The authors declare no conflict of interest.

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
