# Peer review of "Exploring Psychological Needs and Burden of Care in Parents of Children with Hemato-Oncological Diseases"

_hemato, doi:10.3390/hemato3030033_

Round 1

Reviewer 1 Report

Dear Authors,

The subject of this article is very interesting, and it can benefit children with hemato oncological diseases and their parents. The sample is not large, but we all know how difficult it is to enroll patients, parents or caregivers in these studies, as they are particularly overloaded by very tough clinical conditions. My congratulations for the work done.

Below you can read my comments after carefully reviewing your paper:

  • In the Introduction section some issues that have to do with the topic of the article are presented. However, a background of the concepts that are later explained in the objectives of the article is not presented. For example, it does not describe what hemato oncological diseases are, their prevalence in the general population and in children, their life expectancy, etc. Nor is it defined what a parent's adaptation consists of. In line 87 three variables are included in parentheses, but it is not justified why those and not others. If they depend on the measuring instruments, it should be specified. Lastly, what is the child's health-related quality of life is not defined either. In sume, the contents of the Introduction are not linked to the objectives that are presented, so they are not followed in a justified manner.

  • Line 213: There is already scientific literature on how pediatric illness affects the perception of burden of caring parents. In my opinion, the administration of tests to children does not seem to be justified just to extract this relation. Although Table 2 is very complete, and a lot of data is exposed there, the narrative of the article does not go beyond this aspect, and therefore it seems to me that it is not justified to include children in this research. They could, for example, have asked the doctors about the child's illness to obtain data that could influence the parents' perception of overload. That would have been sufficient. From my point of view, the relation between the survey and the objectives of the article is not well established. The link between the queries and these objectives should be justified, in order to legitimate this kind of survey.

  • Of all the data obtained that we can see in Table 2, can we not say anything more than what is described between lines 212 and 221? It would be interesting to  describe these data and to go beyond the caregiver burden concept as we already know.

Reviewer 2 Report

I read the submission entitled "Exploring psychological needs and burden of care in parents of children with hemato-oncological diseases" with great interests because this is an important issue. Caring children with oncology is very hard and it causes high levels of burden and psychological distress. Therefore, I am glad that the authors examined the psychological needs and burden for parents who were caring their children. The authors found out some potential predictors for parents' care burden and proposed some implications. Although the sample size from the perspective of statistics was small, the special population examined in the present study justify the use of a relatively small sample size. Therefore, I believe that the present study's sample size is adequate. However, there are other improvements that should be made and please see my comments below.

1. I think that the abbreviation of QoL should be given in line 88: ...and (d) child's health-related quality of life.

2. I wonder if the authors can report mean and SD for the time since diagnosis.

3. line 109, I cannot understand the meaning of It. ad. in the in-text citation: [24; It. ad. 25].

4. Why do the authors only report Cronbach's alpha for N-DRI but not for CBI and KINDL?

5. Were all the questionnaires tested in Italian?

6. Please clearly mention that the KINDL is assessing generic quality of life. Quality of life can be classified into disease-specific and generic. Therefore, it is important for readers to know which type of quality of life was assessed in the present study.

7. According to the previous comment, please add one more limitation regarding the cancer-specific quality of life. That is, the present study only assessed generic quality of life; therefore, some quality of life items associated with cancer treatments were not examined. This may miss some important information. Future studies are also warranted to use cancer-specific quality of life instrument to corroborate the present study's findings.  

Reviewer 3 Report

The manuscript "Exploring psychological needs and burden of care in parents of 2 children with hemato-oncological diseases" is interesting and relevant. However, there were several issues throughout with typos and a general lack of clarity in the overall writing. Furthermore, the methods were not adequately or correctly described, thus it was difficult to determine the scientific soundness of this work and interpret the results accordingly. Please see below for specific feedback for each section: 

Abstract:

-        First sentence should read “caring FOR a child…”

-        Sentence 2: should read “THE aims…” or “Aims of the study were…”

-        Hodgkin’s disease is spelled incorrectly

-        Overall, a lot of typos and lacking clarity. Can you present stats regarding “Child’s worse physical QoL and parent’s loss of control predicted higher levels of burden”? What was the research design used? How many participants were included?

Introduction:

-        “Following emotional suffering for diagnosis, as for acute lymphoblastic leu-31 kemia [3], parents experience increased caregiver demands, distress, helplessness, and anxiety for future [4] to which it is added uncertainty for a long and invasive treatment protocol [5]” – this sentence is confusing. Please clarify.

-        Overall, some typos and unclear sentences. (e.g. “The assessment of psychological needs and diseases-related problems associated to parental experience of cancer is important to capture factors impacting caregiver burden, but empirical studies are still lacking [21]”). Please revise accordingly.

Methods:

-        2.1 (starting at line 92) – this should describe the methods you used to conduct the study, such as inclusion and exclusion criteria, where participants were recruited from, etc. You have presented the results here, which should be in the results section, not in the methods.

-        Not nearly enough details are presented regarding procedure. There should be enough detail regarding the procedures that would allow the study to potentially be replicated.

-        No details were provided regarding statistical analyses used or how this was done.

-        I suggest using an established checklist for reporting study methods and results. This paper summarizes some options: https://link.springer.com/article/10.1007/s11606-021-06737-1

Results:

-        Because the analyses were not identified in the methods, it’s difficult to understand the results and what was done to obtain them.  

-        Demographic and clinical variables should be presented in this section, not in the methods.

-        Numerous typos throughout this section.

Discussion/conclusion:

-        Several typos and unclear sentences/statements throughout.

-        Some important findings are highlighted, and the authors did a good job highlighting potential limitations and future directions. Qualitative work in this area could also add to/clarify these findings as a potential future direction.

Round 2

Reviewer 1 Report

Dear authors,

The paper has improved significantly. However, I would like to point out one pending question:

The administration of tests to children still does not seem to be justified. The article does not analyze children's results, and therefore it seems to me that it is not still justified to include children in this research. Authors could, for example, have asked doctors about the child's illness to obtain data that could influence the parents' perception of overload. That would have been sufficient. You cited: Litzelman, K.; Catrine, K.; Gangnon, R.; Witt, W.P. Quality of life among parents of children with cancer or brain tumours: 465 the impact of child characteristics and parental psychosocial factors. Quality of Life Research, 2011, 20:8, 1261–1269. And in this paper, they extract child's information from the child's medical record.

Author Response

Dear Reviewer,

we tried to better justify the search for  associations between the child's measures and the parent's measures (lines 107-112). We thinked a multi-dimensional health-related QoL, as detected by KINDL, could be correlated with the parent's burden as detected by CBI. We follow this explorative hypothesis. Well, this correlation emerged between physical well-being and sel-esteem dimensions of KINDL and total burden.

We thank you for your suggestion.

Best regards

Reviewer 3 Report

Dear Authors,

Thank you for the opportunity to review the revised draft of your manuscript. The revisions have significantly improved the overall quality and readability of this manuscript. Please see below for other suggested revisions: 

Introduction:

-        Still some typos and unclear sentences. E.g. lines 90 – 95. This is a very long run on sentence that is confusing. Please be concise.

-        Watch your use of “QOL” throughout. You only need to specify what this means the first time it’s use and can then simply use “QOL” throughout the paper. There’s a lot of going back and forth throughout the manuscript between “QOL” and “quality of life”

-        Line 104: please specify which theoretical model specifically being used/referred to.

Methods:

-        You report 33 children and 44 parents participated. If this was parent-child dyads why aren’t there 44 children as well? Or did some of the parents who participated not necessarily also have children who participated?

-        Lines 124 – 125: Could you please list a few examples of “child’s additional physical illness” that would warrant exclusion.

-        Line 131: “child if younger 7 years old” – this should read “child if younger THAN 7 years old”

-        Lines 149 – 150: “makes it possible to distinguish a serious burding out condition” – do you mean burdening? Please be mindful of typos. There are still several throughout.

-        Thank you for adding in the statistical analysis details. This is very helpful. Please clarify which statistical software you used (e.g. SPSS, R, etc.).

Results:

-        Lines 217 – 218:  “measures did not differentiated as a function of therapeutic conditions too” – do you mean were not differentiated? Did not differentiate? Please watch for typos.

-        Line 219: Should read “revealed NON-independence…”

-        Lines: 231 – 232: “total burden resulted associated with Parental adjustment dimensions” – do you mean resulted in? or were associated with?

Discussion:

-        Line 249: “THE current study…”

-        Lines 262 – 263: “This issue is the only one that resulted associated with higher burden levels…” – resulted in or were associated with? Please read through the discussion carefully! There are still several typos like this.

-        More typos… lines: 179 – 280, 282, 287, 315…

Author Response

Dear Reviewer,

we have accepted all suggestions.

Thank you very much for them.

Best regards